# Selecting the Most Relevant Brain Regions to Classify Children with Developmental Dyslexia and Typical Readers by Using Complex Magnocellular Stimuli and Multiple Kernel Learning

**DOI:** 10.3390/brainsci11060722

**Published:** 2021-05-28

**Authors:** Sara Mascheretti, Denis Peruzzo, Chiara Andreola, Martina Villa, Tommaso Ciceri, Vittoria Trezzi, Cecilia Marino, Filippo Arrigoni

**Affiliations:** 1Child Psychopathology Unit, Scientific Institute, IRCCS Eugenio Medea, 23842 Bosisio Parini, Italy; chiara.andreola@etu.u-paris.fr (C.A.); martina.villa@uconn.edu (M.V.); vittoriatrezzi@gmail.com (V.T.); 2Neuroimaging Lab, Scientific Institute, IRCCS Eugenio Medea, 23842 Bosisio Parini, Italy; denis.peruzzo@lanostrafamiglia.it (D.P.); tommaso.ciceri@lanostrafamiglia.it (T.C.); 3Laboratoire de Psychologie de Développement et de l’Éducation de l’Enfant (LaPsyDÉ), Université de Paris, 75005 Paris, France; 4The Division of Child and Youth Psychiatry at the Centre for Addiction and Mental Health (CAMH), Toronto, ON M6J 1H4, Canada; cecilia.marino@utoronto.ca; 5Department of Psychiatry, University of Toronto, Toronto, ON M5T 1R8, Canada

**Keywords:** developmental dyslexia, fMRI, multiple kernel learning, visual dorsal pathway, attention, dorsal stream vulnerability

## Abstract

Increasing evidence supports the presence of deficits in the visual magnocellular (M) system in developmental dyslexia (DD). The M system is related to the fronto-parietal attentional network. Previous neuroimaging studies have revealed reduced/absent activation within the visual M pathway in DD, but they have failed to characterize the extensive brain network activated by M stimuli. We performed a multivariate pattern analysis on a Region of Interest (ROI) level to differentiate between children with DD and age-matched typical readers (TRs) by combining full-field sinusoidal gratings, controlled for spatial and temporal frequencies and luminance contrast, and a coherent motion (CM) sensitivity task at 6%-CML6, 15%-CML15 and 40%-CML40. ROIs spanning the entire visual dorsal stream and ventral attention network (VAN) had higher discriminative weights and showed higher act1ivation in TRs than in children with DD. Of the two tasks, CM had the greatest weight when classifying TRs and children with DD in most of the ROIs spanning these streams. For the CML6, activation within the right superior parietal cortex positively correlated with reading skills. Our approach highlighted the dorsal stream and the VAN as highly discriminative areas between children with DD and TRs and allowed for a better characterization of the “dorsal stream vulnerability” underlying DD.

## 1. Introduction

Developmental dyslexia (DD) is a complex heritable neurodevelopmental disorder characterized by impaired reading acquisition in spite of adequate neurological and sensorial functioning, educational opportunities, and average intelligence [1]. About 5%–12% of individuals are affected by this disorder, which incurs major lifelong disadvantages in educational and occupational attainment [2].

Reading is a complex task that is driven by neurobiological factors [3,4] and requires the coordination of multiple cognitive and perceptual systems [5,6]. The neurocognitive organization of reading ability depends on rapidly integrating a vast circuit of brain areas over the course of reading skill development. This “reading circuit” is made up of neural systems that support language as well as visual and orthographic processes, working memory, attention, motor movements, and higher-level comprehension and cognition [2,5,7,8,9]. After initial processing of print occurs within the left-hemispheric fusiform gyrus (the visual word form area), a large left hemisphere circuit, including the supramarginal gyrus (orthography to phonology mapping), the superior temporal gyrus (phonological processing), the inferior parietal lobule and the angular gyrus (lexical-semantic processing) and the inferior frontal gyrus (phonological and semantic processing, working memory), is engaged [9,10,11]. Moreover, subcortical regions implicated in long-term and working memory, procedural learning and rapid sequential auditory processing (thalamus, basal ganglia and hippocampus) have also been implicated in reading [12,13,14]. Finally, the left and right fronto-parietal networks [15,16] strongly modulate both the visual and auditory word pathways by temporal and spatial selective attention [17]. As multiple cognitive and sensorial processes are involved in reading, it is probable that a widely variable pattern of weaknesses may contribute to reading difficulties across individuals [18]. While deficits in phonological awareness (i.e., the ability to isolate and manipulate sounds within words) are associated with and may have a causal role in DD [8,19], other theoretical models investigating deficits in underlying sensory (i.e., deficits in the auditory system [20,21,22,23,24,25,26,27,28,29,30]) and cognitive mechanisms (i.e., deficits in the rapid automatized naming [5,31], and in visual and auditory attention [32,33,34,35,36,37,38]) that potentially cause DD to remain compelling.

There is now a great deal of evidence supporting the role of the visual magnocellular (M) system in reading [39]. Reading requires the integration of two distinct neurocognitive systems: a visual system recognizes a visual word from within a crowded group of letters, and a phonological language system rapidly recognizes and produces spoken words from a group of phonemes [40]. The M and parvocellular (P) pathways are major parallel visual system streams that, along with the koniocellular system, constitute the sensory input and process all aspects of the visual world [41,42]. From the retina, the M and P ganglion cells project to different layers in the lateral geniculate nucleus (LGN), constituting the primary thalamic relay of parallel processing between the retina and visual cortex in the mammalian visual system [42,43]. Projections from the LGN remain partly segregated even beyond the primary visual cortex [44], where visual information flows onward along two separate pathways, i.e., the ‘dorsal’ and the ‘ventral’ visual streams, to several areas of the extrastriate visual cortex. In the dorsal stream, visual information flows directly from V1 to V5/MT+ and V6, and runs quite separately from these two areas through two channels, i.e., the ventro-dorsal (involving the inferior parietal lobule) and the dorso-dorsal (involving the superior parietal lobule, SPL) [45]. Visual information from V6 takes two different paths, i.e., the dorsolateral stream, which flows towards V5/MT+ and other areas of the extrastriate visual cortex (i.e., medial superior temporal area, V3A, V4T, ventral part the lateral intraparietal area), and the dorsomedial stream that runs towards the visuomotor areas of SPL (i.e., V6A, medial intraparietal area, ventral intraparietal area) [46]. The dorsal stream is considered blind to colors and responds optimally to contrast differences, low spatial frequencies, high temporal frequencies and real and illusory motion [47,48]. On the contrary, the ventral pathway goes from V1 sublayers 4Cβ, 4A, 3B and 2/3a through V2 and V4 to areas of the inferior temporal lobe (i.e., posterior inferotemporal, central inferotemporal and anterior inferotemporal). This stream is especially associated with stimuli involving color and form, high spatial frequency, low temporal frequency and low luminance contrast [48].

The visual M system contributes to promptly recognizing and sequencing letters. The former happens by rapidly focusing the ventral attention network (VAN) on the letter to be identified, while the latter happens by logging the amplitude and the order of attentional shifts and eye movements during the inspection of each word [32,33,38,49,50]. As the dorsal visual stream is intimately related to attentional systems and particularly to areas shown to be involved in attention control [51,52], the M system plays a vital role in controlling the sequential allocation of attention for reading (attentional shifting) [38,47,53,54,55]. According to the “sluggish attentional shifting” (SAS) hypothesis [32], shifting attention from one object to another is impaired in subjects with DD; is one of the most important predictors of early reading abilities; is a specific target in training that significantly improve reading skills [47]. Therefore, it is plausible to hypothesize that a weakened or abnormal M input in the dorsal visual stream, and the consequent dysfunction of the main fronto-parietal attentional network, could be a neurobiological substrate of SAS in DD [53,54].

Sinusoidal gratings and sensitivity for motion coherence are two classic psychophysical visual tasks [56] that have been widely employed to investigate the neural correlates of the dorsal and ventral visual pathways in the general population [42,43,57,58,59,60,61,62,63,64] and in subjects with DD [44,58,65,66,67], as they tap the psychophysical characteristics of the two cell classes (i.e., M and P) involved in the two visual pathways [68].

Several lines of evidence have shown deficits in the M visual pathway in subjects with DD [69], including abnormally small cells in the M layers of the lateral geniculate nuclei of the thalamus [70,71], impaired perceptual performance [72,73,74,75,76,77,78,79,80,81,82,83] and reduced electrophysiological responses to stimuli mainly processed by the M pathway [53,84,85,86]. However, subjects with DD perform normally on visual tasks preferentially associated with the P pathway [87]. By translating sensory performance into brain functioning, five functional brain imaging studies have revealed reduced or absent activation in areas within the visual M pathway in four independent samples with DD [44,58,65,66,67], although other studies have reported negative findings [88,89]. In particular, Eden and colleagues found no activation in V5/MT during a coherently moving, low-contrast (5%), random-dot stimulus (100% coherence) in eight adults with DD compared to eight typical readers (TRs) [44]. While processing a random-dot stimulus, children with DD showed higher activation in the right inferior and middle frontal gyri (*p* < 0.05 FWE cluster corrected) and reduced activation in the bilateral visual cortex (including V5/MT+; *p* < 0.05 uncorrected) compared to age-matched TRs [58]. Likewise, Olulade and colleagues reported that V5/MT activity was greater for the age-matched TR group than for children with DD (*p*-value = 0.001) during suprathreshold coherent motion (CM) detection (40% coherence) [67]. By using sinusoidal gratings, young adult individuals with DD showed lower activation in both bilateral V1 and MT+ compared to young adult TRs across the full range of contrasts [65,66]. Interestingly, differences in activation within these areas positively correlated with individual differences in reading rate [65,66].

Understanding the role of the visual M pathway in DD and in higher-order cognitive mechanisms related to DD (i.e., attentional shifting) is critical to the early identification and successful treatment of reading disability. Previous fMRI studies have primarily addressed the role of only a few specific brain regions involved in visual motion perception. However, the visual system receives stimulus-driven (bottom-up) as well as a goal-directed (top-down) attentional influence, which modulates all visual processing levels from V1 to visual word form area [15,16,17,38,90]. Investigating this complex pathway presents three challenges. First, previous neuroimaging studies primarily employed univariate analysis to investigate group differences in the primary visual cortex and extrastriate areas. Univariate analysis is the simplest approach for neuroimaging statistical analysis, as it deals with each region (or image voxel) independently from each other. Thus, it investigates just the direct relationship between DD and the single region, without considering the nature of interdependencies between brain regions underlying the visual M stream and the brain regions sensitive to the associated M demands [44,58,65,66,67]. Processing M stimuli activates an extensive brain network that is difficult to characterize when accounting for only a few occipital brain regions. Therefore, the univariate analysis does not enable us to explore the integrated activation of multiple brain regions (i.e., the dorsal visual stream and the attentional systems), nor to investigate group differences. Second, previous imaging studies employed visual tasks controlled for only one functional feature at a time (i.e., spatial or temporal frequencies, or luminance contrast) and often adopted suprathreshold stimuli [44,58,67]. It has been argued that the use of these visual stimuli failed to assess the integrity of the visual streams [42,91]. Previous evidence suggests an advantage in simultaneously manipulating more than one functional feature (e.g., both spatial and temporal frequencies) of the visual stimulus and in using stimuli at both threshold and suprathreshold levels to achieve a better characterization of the visual streams’ responses across the brain [15,16,42,43,64,92,93,94,95,96,97,98,99]. There are two possible interpretations of the threshold in the visual system [100]: (1) the threshold is the minimum level of physical stimulus that can generate output from the sensory process (cf. the high-threshold model) [101]; and (2) the threshold is the physical stimulus that yields a criterion level of performance (cf. the signal detection model) [102]. Third, previous studies have had relatively small samples (adults with DD n = 6 and adult TRs n = 8 [44]; adults with DD n = 5 and adult TRs n = 5 [65,66]; children with DD n = 14, age-matched TRs n = 14 and reading-matched TRs n = 10 [67]), and few neuroimaging studies have directly investigated group differences in the visual M pathway in children with DD [58,67]. These do not provide a reliable account of the mechanisms underlying group differences in visual M stream functioning during development.

To address the above concerns, the present study aimed to investigate whether TRs and children with DD showed different neural activation during two well-established fMRI visual tasks, (1) full-field sinusoidal gratings, in which we simultaneously manipulated spatial and temporal frequencies and luminance contrast close to threshold levels; and (2) sensitivity to motion coherence for both threshold and suprathreshold levels. Whole-brain multivariate analyses implemented through a multiple kernel learning (MKL) machine [103] were used to identify the brain regions sensitive to M stream demands and relevant to the classification task. While univariate methods identify regions especially responsive to different stimulus properties (e.g., low versus high spatial frequency) within the visual M stream, the MKL method assists the execution of a whole-brain multivariate analysis without relying on any a priori assumptions regarding the role of key brain regions in the visual M stream. Thus, this analysis enables the detection of a sparse brain model based on a subset of brain regions that significantly contribute to the visual M stream.

## 2. Materials and Methods

The protocol was approved by the Scientific Review Board and the Ethical Committee of the Scientific Institute, IRCCS Eugenio Medea.

### 2.1. Participants

Twenty-five children with DD (age = 13.92 ± 1.58; 6 females) and 24 TRs (age = 13.13 ± 1.63; 8 females) took part in the present study. Children with DD were recruited from an ongoing project about the genetic basis of DD [104]. Subjects were included if they had a clinical diagnosis of DD [105]. TRs were recruited via 2 different ascertainment schemes: (1) children were contacted by word of mouth among students attending middle and high schools in two districts in northern Italy, i.e., Milan and Lecco; (2) children were selected from a community-based cohort of 819 Italian children aimed at investigating the effects of both genetic and environmental risk factors upon behavioral, cognitive and linguistic measures [106]. For both general population samples, inclusion criteria were: (i) belonging to Caucasian families who were at least first-generation native Italian speakers; (ii) having no certified neurological, neurodevelopmental, visual, hearing, intellectual or motor disabilities; and (iii) having written informed consent signed by both parents.

### 2.2. Neuropsychological Assessment

Both TRs and children with DD underwent the following assessments:(1)IQ, as estimated by the vocabulary and block design subscales of the WISC-III [107];(2)Reading, as assessed by text [108,109], single unrelated words and pseudo-words reading tests [110,111];(3)Verbal working memory (VWM), as assessed by the Single Digit Forward Span, Single Digit Backward Span, Single Letter Forward Span, and Single Letter Backward Span tasks [112];(4)Phonological skills, as assessed by the nonword repetition test (NWR) [113];(5)Hand preference, as assessed by the Briggs and Nebes Inventory (BNI) [114];(6)ADHD traits, as assessed by the Conners’ Parent Rating Scales–Revised:Long version (CPRS-R:L) [115,116,117]. For the current purpose, two subscales were considered: DSM-IV-inattention (DSM-IV-I) and DSM-IV-hyperactivity/impulsivity (DSM-IV-HI).

To be included, both TRs and children with DD were required to have a mean score between vocabulary and block design subtests of the WISC-III of ≥7 (i.e., ≥−1.00 SD) [107], no other neuropsychiatric diagnoses, no major contraindications to MRI, and normal or corrected-to-normal vision. Children with DD were included if they had either accuracy or speed z-score of ≤−2.00 SDs on text or single unrelated words or pronounceable pseudo-words reading tests. TRs were included if they had both accuracy and speed z-scores of ≥−1.00 SD on all reading tests.

Table 1 shows the descriptive statistics of demographic and neuropsychological variables of both DD and TR groups after the fMRI data quality check (cf. ‘2.6 fMRI data processing’ paragraph). As expected, children with DD showed significantly lower scores in all reading tests compared to TRs. To control for confounders, block design and DSM-IV-I were entered as covariates in subsequent analyses of imaging data processing (cf. ‘2.6 fMRI data processing’ paragraph). Regarding IQ, we decided to control only for block design because several previous cross-sectional and longitudinal studies reported a relationship between reading skills and verbal IQ [118,119,120,121,122,123]. Moreover, as mean bivariate correlations (r) were substantial within reading and VWM tests (reading *r* = 0.766, Appendix A; VWM *r* = 0.391, Appendix A), we created two composites by averaging each task within reading and VWM.

### 2.3. MRI Acquisition Protocol

MRI data were acquired on a 3T Philips Achieva d-Stream scanner (Best, The Netherlands) with a 32-channel head coil. Visual stimuli were developed with Presentation^®^ software (Neurobehavioral System Inc., Berkeley, CA, USA) and delivered through a VisuaStim digital device for fMRI (Resonance Technology Inc., Northridge, CA, USA). MRI-compatible goggles with two displays were used, with a 60 Hz frame rate and 800 × 600 spatial resolution (4/3 aspect ratio) subtending a horizontal visual angle of 30°. An MRI-compatible pad was used to record subjects’ answers and response times. The MRI protocol included the use of an anatomical T1-weighted (T1W) 3D Turbo Field Echo sequence as a subject morphological reference of MRI data (Field Of View (FOV) = 256 × 256 × 175 mm^3^, voxel size 1 × 1 × 1 mm^3^, Time of Repetition (TR) = shortest (~8.1 ms), Time of Echo (TE) = shortest (~3.7 ms), Flip Angle (FA) = 8°). The fMRI data were acquired with a T2*-weighted Gradient Echo planar sequence (FOV = 240 × 240 mm^2^, voxel size = 3 × 3 mm^2^, slice thickness = 3 mm, slice gap = 0.5 mm, slice number = 39, TR = 2 s, TE = 26 ms, FA = 90°).

### 2.4. fMRI Task Design

#### 2.4.1. Full-Field Sinusoidal Gratings

The task consisted of 14 s blocks of “M stimuli,” “P stimuli” and blank stimuli (fixation point only). The M and P stimuli were designed to elicit differential bold responses from M and P pathways [43]. The M stimulus was a monochrome, low spatial frequency, high temporal frequency, high luminance contrast, full-field sinusoidal grating with sinusoidal counterphase flicker; the P stimulus was a high color contrast, high spatial frequency, low temporal frequency, low luminance contrast full-field sinusoidal grating with sinusoidal counterphase flicker. The M stimulus was a 100% luminance contrast, black–white grating with a spatial frequency of 0.5 cycles per degree (cpd) and a flicker frequency of 15 Hz. The P stimulus was a low luminance contrast, high color contrast red–green grating with a spatial frequency of 2 cpd and a flicker frequency of 5 Hz. Color levels in the P stimulus were set to be near-isoluminant, the red luminance was set to the maximum level, and the green was set to 39% of the maximum level, as implemented in Denison and colleagues [43]. The blank stimulus was a gray screen of mean luminance. The outer borders of each stimulus faded into gray to avoid sharp visual edges at the stimulus boundaries. Both gratings were presented at one of 6 orientations (0°, 30°, 60°, 90°, 120° and 150°) and changed to the next orientation every 2.33 s. The protocol included 28 blocks (8 M, 8 P and 12 blank) presented in pseudorandom order with the constraint that the same stimulus type could not appear in adjacent blocks to minimize adaptation to the stimuli. A white fixation point subtending 0.2° visual angle appeared at the center of the screen throughout the stimulus blocks. Subjects were instructed to maintain fixation throughout the run, and they performed an irrelevant target detection task during the M and P stimulus blocks to encourage them to do so. The target was a bidimensional Gaussian contrast reduction patch. Its size was linearly scaled with the distance from the fixation point. The target appeared for 300 ms at random times and in random positions 50% of the time during the second half of each stimulus block. At the end of each stimulus block, the screen turned gray, and subjects were asked to press the corresponding button in the response pad to answer questions (i.e., “Did the target appear?”—right button for “Yes” and left button for “No”). Subjects had 4s to answer the question. There was a 2s inter-stimulus waiting period between stimulus blocks.

#### 2.4.2. CM Detection

Sensitivity to motion coherence was assessed for radial motion (expanding or contracting), which has previously proved able to evoke more activation than simple coherent motion (i.e., vertical/horizontal) [57,125], and which made it easy for subjects to maintain fixation [95]. The stimuli comprised 50 small white and 50 small black dots (each 20 arcmins), presented for 250 ms on a mean luminance gray background. A proportion of dots drifted coherently at a speed of 10°/s (limited lifetime of 8 frames, frame rate 60 Hz), while the remainder were displayed in random positions on each frame. According to previous studies on children with and without neurodevelopmental disorders [63,67,93,94,125,126], we used three levels of coherently moving dots (CML: Coherent Motion Level), i.e., 6%, 15% and 40%. At the beginning of each stimulation block, a white fixation point subtending 0.2° visual angle appeared at the center of the screen for 0.5 s and was followed by the 0.25 s CM stimulus. Subjects were instructed to maintain fixation throughout the run, and were actively engaged in performing a motion detection task and pressing the corresponding button on the response pad to answer questions (i.e., right button for expanding and left button for contracting). After the stimulus, subjects had 4 s to answer the question and were asked to give an answer even when they could not detect the motion direction. There was a 4.25 s inter-stimulus waiting period between stimulus-blocks. The protocol included 48 stimuli (8 repetitions for each combination of coherence level and motion direction) administered in a pseudorandom order with the constraint that the same coherence level could not appear in more than two adjacent blocks regardless of the motion direction.

### 2.5. Anatomical MRI Data Analysis

T1W images were corrected for bias field intensity artifacts using the N4 algorithm [127]. Subsequently, FreeSurfer tools (http://surfer.nmr.mgh.harvard.edu/, version 6.0) were used to further process the T1W images following the recon-all processing pipeline. The HCP-MMP1 atlas was used to divide each hemisphere into 180 regions, which can be grouped into 22 macro-regions [128].

### 2.6. fMRI Data Processing

The fMRI data were processed following the FreeSurfer Functional Analysis Stream (FSFAST, version 6.0). The preprocessing pipeline included motion correction, slice-timing correction, resampling on the ‘fsaverage’ template, smoothing, and intensity normalization. Template resampling was performed by exploiting the subject T1W images as an intermediate step, and smoothing was performed using a 3 mm FWHM filter.

Outlier volume detection was performed using an ad-hoc software we made by selecting all the volumes with (1) an overall motion from the previous volume larger than 2 mm, or (2) a mean intensity difference from the previous volume larger than 2.5 times the standard deviation in the whole run. The fMRI runs were excluded from the study if more than 20% of their overall volume was tagged as an outlier or if more than 30% of their volume was tagged as an outlier for a single stimulus condition. This led to the exclusion of three children with DD (all males) and two TRs (one male). The excluded subjects did not differ from the included ones in any of the selected demographic or neuropsychological variables (data available upon request). The final groups had 22 subjects each. The number of outlier volumes for each task of each included subject did not significantly differ between the groups (full-field sinusoidal gratings, *p*-value = 0.34; CM detection, *p*-value = 0.42).

For each subject, a first-level analysis was performed with a GLM model using the task conditions as predictors of interest and the motion parameters and outlier volumes as nuisance predictors. Subsequently, contrast maps were defined within each task. Regarding the full-field sinusoidal gratings, two contrast maps were specified, i.e., M stimulus vs. Baseline (M-*vs*-B) and P stimulus vs. Baseline (P-*vs*-B). Three contrast maps were outlined for the coherent motion, one for each level of motion coherence, i.e., Coherent Motion Level 6% vs. Baseline (CML6-*vs*-B), CML15-*vs*-B, CML40-*vs*-B. Finally, each contrast map was partitioned into Regions of Interest (ROI), which is an approach that extracts data from a subset of voxels belonging to a homologous anatomo-functional brain region, improving the signal-to-noise ratio. In particular, the mean value of each contrast map was computed for each subject in the cortical ROI from the HCP-MMP1 atlas by using a linear model that allowed to control for the effects of performance IQ and DSM-IV-I in order to remove the amount of the signal explained by these confounders from our dependent variable (i.e., the mean value of each contrast map) (see Appendix A for a flowchart of the pipeline).

### 2.7. Multivariate Analyses

A machine learning-based analysis was performed on the ROI data (controlled for the effects of performance IQ and DSM-IV-I) to investigate whether visual fMRI tasks can be used to discriminate between TRs and children with DD and to detect which ROIs contribute most to the classification.

In the classification experiments, we used a multiple kernel technique [129,130] based on the Support Vector Machine (SVM) model to combine multiple contrast maps while preserving the topological information (see Appendix A for a flowchart of the classification experiment). In the SVM model, we included a weight for each kernel learned during the training (higher weights were assigned to the kernels providing the largest discriminative information). As a different linear kernel was associated with each ROI, the kernel weights quantified the contribution of each ROI to the final classification and could be interpreted as ROI weights. Moreover, for each ROI, the mean values from the different contrast maps were concatenated in a single input vector, thus preserving the topological information among the tasks. In this study, we employed the Group Lasso–Multiple Kernel Learning algorithm (GL–MKL) [131], which included a sparsity contribution to the regulation term during the kernel weights training procedure. The sparsity parameter (p) and the SVM error penalty parameter (C) were estimated by adopting a grid search approach and a double cross-validation procedure to avoid overfitting. More precisely, in each iteration of the outer cross-validation cycle, two subjects, one DD and one TR, were used as a testing set to avoid the anticorrelation effect with the training set [132]. In the inner cross-validation cycle, a ten-fold cross-validation procedure was used to optimize the classifier parameters. Classifier performances were evaluated using classification accuracy, the area under the ROC curve (AUC) and the *p*-value. The *p*-value was computed using a permutation test with 10,000 permutations of the subject labels.

Theoretically, all ROIs contributed to the classification task; thus, we performed an analysis to identify the subset of ROIs that contributed the most to the classification. Following a Greedy backward elimination approach [133], we fixed the model parameters (C, *p*-value) to the consensus values derived from the cross-validation procedure, and we iteratively backward removed the ROI with the lowest weight and retrained the classifier. We selected the configuration with the best performance (i.e., accuracy, AUC) and extracted the set of ROIs that best differentiated between TRs and children with DD. As the magnitude of the SVM weights associated with each feature did not have a direct neurophysiological interpretation, linear backward models were transformed into forward models [134]. In each forward model, a weight (contrast weight) was associated with each element of the input vector (i.e., the ROI contrast values) and could be interpreted as generative models (e.g., GLM). Contrast weights indicated the contribution of the target class (i.e., DD) to each element of the feature vector (i.e., the value of each contrast map activation in the given ROI) and were used to identify which task held the largest discriminative information within each ROI.

As contrast weight analysis is influenced by the complementary information provided by the different contrasts in the selected ROIs, we performed univariate post hoc analyses (i.e., *t*-tests) to independently investigate the direction and magnitude of the differences between TRs and children with DD for each contrast.

Correlations between the mean activation of each contrast map (i.e., M-*vs*-B, P-*vs*-B, CML6-*vs*-B, CML15-*vs*-B and CML40-*vs*-B) within the significant ROIs and the neuropsychological domains (i.e., reading, VWM and phonology) in the total sample were calculated using Pearson correlations as implemented in IBM SPSS Statistics for Windows, Version 21.0 (IBM Corp. Released 2012).

To summarize, the whole multivariate analysis can be broken down into four steps: firstly, we performed a classification experiment to discriminate between TRs and children with DD by using whole-brain ROIs. Secondly, we performed an analysis on the selected classifier to identify which ROIs and contrast maps contributed most to the classification. Thirdly, we ran univariate post hoc analyses (i.e., *t*-tests) to investigate the direction and the magnitude of the differences between TRs and children with DD in the selected ROIs. Finally, we tested for correlations between task-induced cortical activation in the different ROIs and the neuropsychological tests.

## 3. Results

### Multivariate Analyses—The Group Lasso–Multiple Kernel Learning Algorithm

Using the GL–MKL on the HCP-MMP1-defined ROIs for all contrast maps resulted in a model which discriminates between TRs and children with DD with 65.9% accuracy and 64.7% AUC (*p*-value = 0.043). On the basis of the post hoc ROI weight analysis (see “Section 2.7”), we identified a set of 11 ROIs that were likely to have contributed most to the classification (Figure 1).

Figure 2 shows the ranking of the selected ROIs and the corresponding ROI-weight.

Table 2 reports the ROI weights and the contrast weights obtained with the forward model. Univariate post hoc statistical analyses at the ROI level for each contrast are reported in Table 3. Although none of the *t*-tests survived a whole-brain multiple comparison correction, the qualitative interpretation of the *t*-values provides useful information about the group differences, which are complementary to the significant results obtained with the multivariate analysis. Almost all *t*-values were positive, indicating that activation in discriminative ROIs is generally higher in TRs than in children with DD (Table 3). Furthermore, the magnitude of the *t*-values for threshold CM contrasts (i.e., CML6-*vs*-B and CML15-*vs*-B) and the M-*vs*-B contrast showed on average larger differences between the two groups compared to the other contrasts (Table 3). Interestingly, the ROIs with the largest differences between the two groups show higher contrast weights also in the multivariate analysis (Table 2). 

Several nominally significant correlations were found between individual differences in the mean activation of the contrast maps in the significant ROIs and reading, VWM and phonology (see Appendix A). However, after applying the Bonferroni correction for multiple testing (threshold to infer statistical significance of *p*-value = 0.0003; 11 ROIs for five contrast maps for three neuropsychological domains), only the correlations between the mean activation of CML6-*vs*-B within the right lateral area 7P and the right medial area 7A, and reading survived (*r* = 0.552, *p*-value = 0.0001, and *r* = 0.540, *p*-value = 0.0002, respectively) (Figure 3). Specifically, a mean hyperactivation in these ROIs correlated with better performance in reading skills.

## 4. Discussion

To the best of our knowledge, the current study is the first to present an MKL-based methodology to differentiate groups of children with DD and age-matched TRs by using two visual tasks, one at both threshold and suprathreshold levels (sensitivity to motion coherence at 6%, 15% and 40%) and one in which more than one functional feature is simultaneously manipulated (full-field sinusoidal gratings controlled for both spatial and temporal frequencies and luminance contrast close to threshold levels). The multivariate approach performed by using the MKL-ROI classifier allows us to explore the widespread effects of the diagnosis of DD across multiple brain ROIs. Thus, it is more suitable to investigate the neural correlates of neurodevelopmental disorders involving wide brain networks compared to univariate analysis, aimed to test the relationship between the diagnosis of DD and every single ROI.

Overall, the multivariate approach significantly discriminated between TRs and children with DD. In particular, we demonstrated that functional activation in the entire visual dorsal stream and the VAN ranks highest (Table 2). Although none of the T-test would survive the multiple comparison correction, their qualitative analysis indicated that activation in discriminative ROIs is generally higher in TRs than in children with DD (Table 3) [44,58,65,66,67]. This pattern of results is consistent with the notions that the fronto-parietal network is linked to “dorsal stream vulnerability” underlying several neurodevelopmental disorders and that the functionality of the M pathway is intimately related to the attention systems involved in attention control [51,52]. Deficits in the M pathway could influence higher visual processing stages through the dorsal stream and, therefore, lead to reading difficulties through impaired attentional orienting [32,38,69]. While the right fronto-parietal system is a crucial component of the network subserving the automatic shifting of attention [15,16], the left fronto-parietal system has been linked to auditory word form processing [135]. Developmental changes in the activation of the right fronto-parietal system have been linked to reading skills in both children with DD [136,137,138] and TRs [139]. Taken together, our findings suggest that a weakened or abnormal M input in the dorsal visual stream, and consequent dysfunction of the main fronto-parietal attentional network, are associated with SAS in DD [53,54].

Among the contrasts that influenced the classification algorithm most, functional activations induced by CM have a higher weight in the MKL classifier (Table 2). Although the contrast weights across the different levels of coherence are comparable within these ROIs, comparison among the coherence levels could be relevant to discriminate between TRs and children with DD. Consistently, the qualitative interpretation of the contrast T-values for CM at threshold levels (i.e., 6% and 15%) suggested that they are often higher in magnitude compared to the contrast weights for CM at suprathreshold level (i.e., 40%) (Table 3). The CML6 and CML15 showed the most group-discriminating activations in extra-visual areas lying within the dorsal portion of the M stream, the VAN and the salient network (Table 2) [15]. According to findings reported by independent psychophysical studies [93,94,125,126,140], we can hypothesize that the CML6 and CML15 represent relevant sensory stimuli requiring additional attentional resources to appropriately process the visual stimulus. According to this view, low CM levels might work as a ‘circuit breaker’ for the top-down attentional network, and they might imply the override of the current attentional set—usually activated by a high CM level—leading to the engagement of the frontal areas to enhance the attentional resources.

After a qualitative interpretation of the data, it is important to note that although children with DD showed lower activation compared to TRs for both the CML6-*vs*-B and the CML15-*vs*-B contrast maps, the magnitude of T for the CML6-*vs*-B is greater than those for the CML15-*vs*-B. Moreover, the mean activation of the CML6-*vs*-B within some areas of the right superior parietal cortex (i.e., the lateral area 7P and the medial area 7A) significantly correlated with reading (Appendix A). These findings are consistent with the SAS hypothesis [32] and the “perceptual noise exclusion deficit” [126,140]. Before the letter-to-sound mapping mechanism is applied, irrelevant lateral letters should be filtered out by accurate and rapid shifts in spatial and temporal visual attention [141,142,143,144,145,146,147,148]. This process may be more difficult if visual processing is hampered by deficits in attentional shifting and noise exclusion. Thus, the role of an auditory and phonological disorder aside, visual attention shifting and noise exclusion play a critical role in letter-to-speech sound integration during letter string processing because they are crucially involved in forming representations enabling efficient recognition of letters and letter sequences, identification of word shape and boundaries between words, representation of the sequential orthographic structure and the development of phonological representations [38,126,140].

Regarding the full-field sinusoidal gratings, the ROIs that lead the classification in this task have the lowest ROI weights in the MKL classifier (Table 2). Although the M-*vs*-B and P-*vs*-B contrast weights are comparable within these ROIs, the M-*vs*-B T-values are often higher in magnitude compared to the P-*vs*-B (Table 3). So, as has been hypothesized for the CM, it is likely that the comparison between M and P stimuli could be relevant to the classification of TRs and children with DD. These findings further suggest that the M pathway is impaired in individuals with DD, whereas the other major parallel pathway of the visual system, the parvocellular stream, is less severely or not at all affected [38,39,54,69].

Limitations of the present investigation must be acknowledged. First, although we tested a larger sample size than those described in previous studies, the sample size of our study is still relatively small, which may have affected the classification performance and model generalizability [149]. Nonetheless, our sample was sufficient for a ranking profile of brain regions consistent with previous neuroimaging findings and etiological models of DD. It is plausible to predict that the greater power afforded by using larger samples would produce classification accuracy indices larger than those reported herein. Second, our results cannot be generalized to the general population since our aim was to use the MKL-based methodology as an alternative and stronger approach to investigate the role of the M pathway in DD and not to foster the diagnostic process. Third, the implemented procedure selects the minimal number of ROIs that provide a significant contribution to the classification. Consequently, ROIs that give a minor contribution to the classification but are involved in the “reading network” may not be highlighted.

## 5. Conclusions

The MKL-ROI approach used in the present study identified a highly discriminative network in the entire visual dorsal stream and the VAN when comparing children with DD and age-matched TRs. These brain areas lie within the fronto-parietal network, which is linked to the “dorsal stream vulnerability” underlying several neurodevelopmental disorders, and are known to be involved in deficits in the SAS in DD. Moreover, the implementation of both threshold- and suprathreshold-level visual stimuli in which more than one functional feature is simultaneously manipulated allowed us to better characterize the “dorsal stream vulnerability” underlying DD. According to the M theory of DD [38,54,69], a weakened or abnormal M input in the dorsal visual stream may lead to dysfunction of the main fronto-parietal attentional network [53,54] and difficulties in noise exclusion [150]. Our results further support the M visual pathway as a reliable biomarker of DD [39,151], which could lead to new approaches in the diagnosis of this neurodevelopmental disorder [152]. Moreover, these findings pave the way for the creation of early identification protocols, more effective prevention programs and better-defined rehabilitative treatments. Based on previous findings [35,47,153], it is plausible to assume that treatment programs based on M stream training could be a new approach for remedying and even preventing DD.

## Figures and Tables

**Figure 1 brainsci-11-00722-f001:**
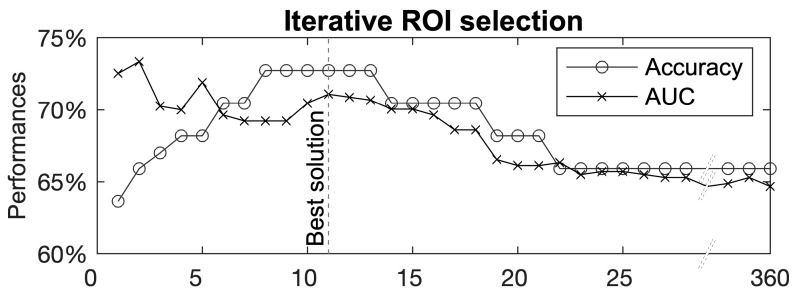
Post hoc analysis to select the subset of ROIs that contributed most to the classification. The *X*-axis reports the number of ROIs used for the classification. The *Y*-axis reports classification performance scored with that set of ROIs. ROIs are iteratively removed from the lowest to the highest weight, which was assigned by assessing all of them. The red dashed line indicates the best configuration.

**Figure 2 brainsci-11-00722-f002:**
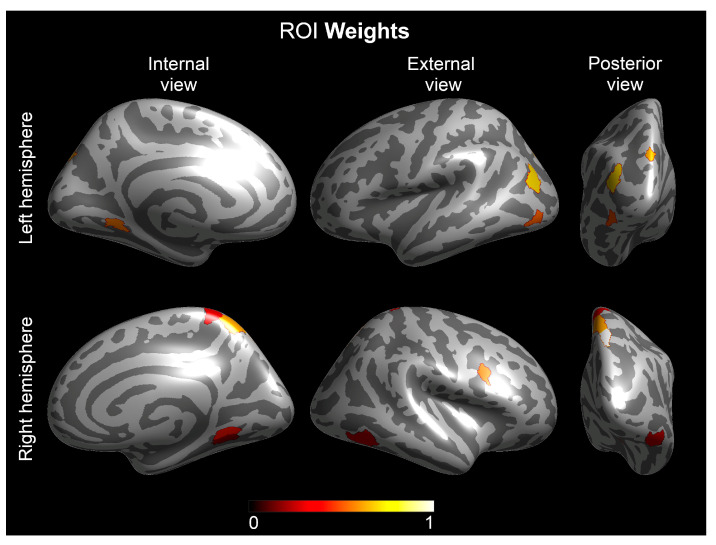
ROI weights for the 11 ROIs that provide the highest GL–MKL classifier performance. ROIs from the HCP-MMP1 atlas [128] were selected by the MKL to classify TRs and children with DD. ROI weights were scaled to a maximum value equal to 1, and the color of the regions varies from black (minimum ROI weight) to white (maximum ROI weight).

**Figure 3 brainsci-11-00722-f003:**
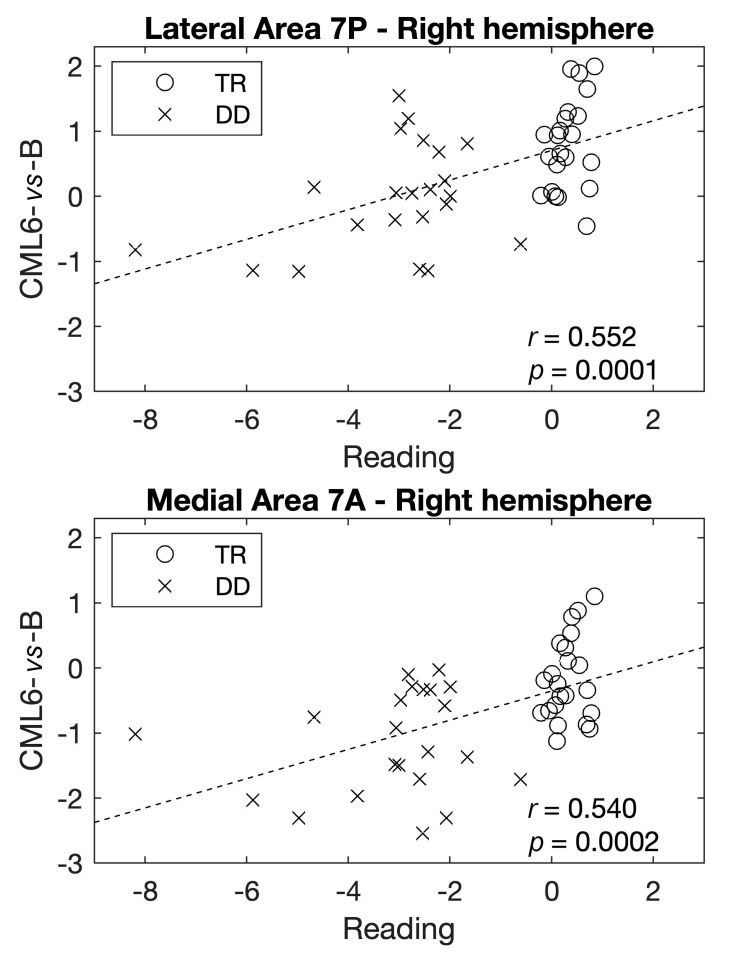
Bonferroni significant correlations between the mean activation of contrast maps within the significant ROIs and the neuropsychological domains in the total sample (n = 44). (**a**) Correlation between the mean activation of CML6-*vs*-B within the right lateral area 7P and reading. (**b**) Correlation between the mean activation of CML6-*vs*-B within the right medial area 7A and reading. Reading represents a mean score of text [98,99], single unrelated words and pronounceable pseudo-words [100,104] reading tests.

**Table 1 brainsci-11-00722-t001:** Descriptive statistics of demographic and neuropsychological variables.

		Children with DD (n = 22)					Typical Readers (n = 22)					Χ^2^	df	*p*
**Sex (Male/Female)**		16/6					15/7					0.11	1	0.741
**Handedness**	**Right-handed**	18					17					0.31	2	0.856
	**Left-handed**	3					3							
	**Ambidextrous**	1					2							
		**Min**	**Max**	**Mean (SD)**	**Skewness**	**Kurtosis**	**Min**	**Max**	**Mean (SD)**	**Skewness**	**Kurtosis**	***t*-Test**	**df**	***p***
**Age**		10.92	16.5	14.10 (1.48)	−0.40	−0.22	11	16.25	13.18 (1.65)	0.47	−1.31	−1.94	42	0.059
**IQ ^†^**		7.0	13.5	10.59 (1.78)	−0.10	−0.61	7.5	18.5	13.70 (2.82)	−0.67	0.23	4.38	42	<0.001
**IQ, Vocabulary**		5.0	16.0	9.73 (2.75)	0.25	−0.24	8.0	18.0	13.23 (3.09)	−0.27	−0.95	3.98	42	<0.001
**IQ, Block Design**		8.0	18.0	11.82 (2.04)	1.16	3.10	7.0	19.0	14.18 (3.65)	−0.47	−0.49	2.65	42	0.012
**TR, accuracy**		−10.43	0.74	−3.34 (2.42)	−1.30	2.52	−0.13	1.42	0.61 (0.34)	0.10	0.88	7.61	42	<0.001
**TR, speed**		−4.21	−0.09	−2.55 (1.01)	0.44	−0.11	−1.00	1.18	0.16 (0.65)	0.02	−1.00	10.55	42	<0.001
**SWR, accuracy**		−10.00	0.33	−3.41 (2.63)	−0.74	0.22	−0.67	1.00	0.27 (0.56)	−0.04	−1.16	6.42	42	<0.001
**SWR, speed**		−10.12	−0.41	−3.66 (2.16)	−1.49	2.77	−1.11	0.87	0.04 (0.57)	−0.19	−0.96	7.75	42	<0.001
**SPWR, accuracy**		−8.50	0.33	−2.28 (1.91)	−1.57	4.33	−0.67	1.33	0.50 (0.46)	−0.47	0.60	6.61	42	<0.001
**SPWR, speed**		−9.42	−0.67	−3.38 (2.42)	−1.43	1.36	−1.03	1.40	0.26 (0.64)	−0.26	−0.65	6.82	42	<0.001
**SLFS**		−2.30	0.65	−1.15 (0.72)	0.46	0.61	−1.35	1.35	0.14 (0.76)	−0.69	−0.28	5.72	41	<0.001
**SLBS**		−2.20	1.35	−0.71 (0.75)	0.93	2.25	−1.35	1.60	−0.07 (0.89)	0.73	−0.41	2.53	41	0.015
**SDFS**		−2.20	0.00	−1.38 (0.62)	0.49	−0.08	−2.00	1.00	−0.61 (0.74)	0.18	−0.21	3.68	41	0.001
**SDBS**		−1.35	0.65	−0.56 (0.47)	0.56	1.01	−1.40	2.00	0.10 (0.96)	0.91	−0.20	2.84	41	0.007
**SNWR**		−7.00	3.53	−1.66 (2.71)	−0.1	−0.41	−2.79	3	1.18 (1.40)	−1.45	2.26	4.38	42	<0.001
**ADHD**	**DSM-IV-I ^‡^**	41	82	60.09 (10.73)	0.33	−0.32	39	59	46.11 (5.49)	0.94	0.34	−5.43	42	<0.001
	**DSM-IV-HI ^§^**	38	71	49.20 (7.96)	1.13	1.57	38	65	47.36 (7.63)	1.099	0.377	−0.76	40	0.450
**SES ^¶^**		20	90	58.84 (19.16)	−0.08	−0.37	30	90	60.23 (19.42)	0.19	−1.04	0.56	39	0.579

TR: Text reading; SWR: single words reading; SPWR: single pseudo-words reading; SLFS: single letters forward span; SLBS: single letters backward span; SDFS: single digits forward span; SDBS: single digits backward span; SNWR: single non-word repetition. ^†^ Mean score of vocabulary and block design subtests of the WISC-III [108]. ^‡^ The DSM-IV-Inattention (DSM-IV-I) subscale of the CPRS-R:L [116,117,118]. ^§^ The DSM-IV-hyperactivity/impulsivity (DSM-IV-HI) subscale of the CPRS-R:L [116,117,118]. ^¶^ As estimated by father’s/mother’s employment [124].

**Table 2 brainsci-11-00722-t002:** List of the ROIs with the highest performance.

	Hemisphere	ROI ^†^	ROI-Region ^†^	ROI Weight	Contrast Weight				
					M-*vs*-B	P-*vs*-B	CML6-*vs*-B	CML15-*vs*-B	CML40-*vs*-B
**1**	Right	Lateral Area 7P	Superior Parietal Cortex	1.000	−0.381	−0.277	−0.583	−0.525	−0.404
**2**	Left	Area PGp	Inferior Parietal Cortex	0.704	−0.405	−0.298	−0.613	−0.491	−0.361
**3**	Left	Area V6A	Dorsal Stream Visual Cortex	0.644	−0.407	−0.066	−0.549	−0.693	−0.220
**4**	Right	Medial Area 7A	Superior Parietal Cortex	0.633	−0.209	−0.257	−0.759	−0.458	−0.324
**5**	Left	Ventro-Medial Visual Area 1	Ventral Stream Visual Cortex	0.585	−0.112	0.119	−0.524	−0.664	−0.507
**6**	Left	Area Lateral Occipital 2	MT+ Complex and Neighboring Visual Area	0.539	−0.388	0.157	−0.495	−0.483	−0.589
**7**	Right	Area IFJ posterior	Inferior Frontal Cortex	0.521	−0.685	−0.477	−0.500	−0.230	−0.033
**8**	Right	Ventro-Medial Visual Area 1	Ventral Stream Visual Cortex	0.436	−0.546	−0.243	−0.436	−0.497	−0.454
**9**	Right	Area 5-L	Paracentral Lobular and Mid-Cingulate Cortex	0.315	−0.060	−0.077	−0.773	−0.524	−0.344
**10**	Right	Area PH	MT+ complex—Ventral stream fusiform face complex	0.290	−0.561	−0.573	−0.321	−0.424	−0.272
**11**	Right	Ventro-Medial Visual Area 2	Ventral Stream Visual Cortex	0.222	−0.655	−0.369	−0.396	−0.409	−0.333

^†^ As parceled out in the HCP-MMP1 atlas [129]; M-*vs*-B: M stimulus vs. Baseline; P-*vs*-B: P stimulus vs. Baseline; CML6-*vs*-B: Coherent Motion Level 6% vs. Baseline; CML15-*vs*-B: Coherent Motion Level 15% vs. Baseline; CML40-*vs*-B: Coherent Motion Level 40% vs. Baseline.

**Table 3 brainsci-11-00722-t003:** Group differences (*t*-tests) within each contrast map in the highest performance ROIs.

	Hemisphere	ROI ^†^	ROI-Region ^†^	M-*vs*-B	P-*vs*-B	CML6-*vs*-B	CML15-*vs*-B	CML40-*vs*-B
**1**	Right	Lateral Area 7P	Superior Parietal Cortex	2.309 (0.026)	1.713 (0.094)	3.662 (0.001)	2.993 (0.005)	2.009 (0.051)
**2**	Left	Area PGp	Inferior Parietal Cortex	2.023 (0.049)	1.770 (0.084)	2.985 (0.005)	2.503 (0.016)	1.579 (0.122)
**3**	Left	Area V6A	Dorsal Stream Visual Cortex	1.324 (0.193)	0.262 (0.795)	2.230 (0.031)	2.342 (0.024)	0.866 (0.391)
**4**	Right	Medial Area 7A	Superior Parietal Cortex	1.131 (0.264)	1.327 (0.192)	4.497 (<0.001)	2.413 (0.020)	1.916 (0.062)
**5**	Left	Ventro-Medial Visual Area 1	Ventral Stream Visual Cortex	0.839 (0.406)	−0.751 (0.457)	2.481 (0.017)	2.906 (0.006)	2.289 (0.027)
**6**	Left	Area Lateral Occipital 2	MT+ Complex and Neighboring Visual Area	1.805 (0.078)	−0.615 (0.542)	2.151 (0.037)	2.026 (0.049)	2.727 (0.009)
**7**	Right	Area IFJ posterior	Inferior Frontal Cortex	3.948 (<0.001)	2.468 (0.018)	2.698 (0.010)	1.430 (0.160)	0.194 (0.847)
**8**	Right	Ventro-Medial Visual Area 1	Ventral Stream Visual Cortex	3.014 (0.004)	1.288 (0.205)	1.845 (0.072)	1.905 (0.064)	1.810 (0.077)
**9**	Right	Area 5-L	Paracentral Lobular and Mid-Cingulate Cortex	0.312 (0.756)	0.316 (0.754)	4.441 (<0.001)	2.509 (0.016)	1.642 (0.108)
**10**	Right	Area PH	MT+ complex—Ventral stream fusiform face complex	3.538 (0.001)	3.078 (0.004)	1.553 (0.128)	2.110 (0.041)	1.401 (0.169)
**11**	Right	Ventro-Medial Visual Area 2	Ventral Stream Visual Cortex	2.628 (0.012)	1.437 (0.158)	1.467 (0.150)	1.789 (0.081)	1.148 (0.257)

^†^ As parceled out in the HCP-MMP1 atlas [129] M-*vs*-B: M stimulus vs. Baseline; P-*vs*-B: P stimulus vs. Baseline; CML6-*vs*-B: Coherent Motion Level 6% vs. Baseline; CML15-*vs*-B: Coherent Motion Level 15% vs. Baseline; CML40-*vs*-B: Coherent Motion Level 40% vs. Baseline. Uncorrected *p*-values are reported in parentheses. NOTE: none of the significant differences survived after correction for multiple comparisons. Positive *t*-values indicate higher activations in TR than in children with DD.

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
