# Peer review of "Selecting the Most Relevant Brain Regions to Classify Children with Developmental Dyslexia and Typical Readers by Using Complex Magnocellular Stimuli and Multiple Kernel Learning"

_brainsci, 2021, doi:10.3390/brainsci11060722_

Round 1
Reviewer 1 Report
This study aims to characterize the brain networks that may be related to a deficit in the Magnocellular (M) system in dyslexic readers. To this end the authors employed stimuli that are known to activate dorsal and ventral visual streams which receive input from the M system and Parvocellular (P) systems respectively, using two tasks 1) full-field sinusoidal gratings and 2) motion coherence. They manipulated spatial and temporal frequencies, luminance contrast, as well as threshold and supra-threshold levels, in an attempt to design more sensitive tasks that assess the integrity of the dorsal and ventral visual streams within the M and P visual systems. While previous studies have employed univariate analyses to examine group differences in visual processing (between Dyslexic and typical readers), the current study employed whole-brain multivariate analyses to identify disparate attentional networks in the brain that may contribute to the visual M stream. They found that weaker neural activation along the dorsal visual stream and in areas within the fronto-parietal attentional network in dyslexic readers was able to discriminate between Dyslexic and typical reading controls.
Overall, the manuscript is well-written. The authors outline the theoretical and methodological motivations for their study well, they highlight how visual processing within the M system relates to reading specific processes, and they draw clear conclusions from their findings. The results contribute to research geared towards early identification of Developmental Dyslexia and possible approaches to remediation that focus on perceptual training targeting the dorsal visual stream.
Major Comments:
- It would help readers to follow the argument of the manuscript if the magnocellular (M) system and dorsal and ventral visual streams were defined earlier in the introduction because the second paragraph is not clear. In the current version, it was also not clear that the dorsal and ventral streams receive different input from the M and P visual systems. I recommend reordering the structure of the second and third paragraphs as follows:
- Describe the two neurocognitive systems involved in reading.
- Define the M and P systems, and dorsal and ventral streams.
- Discuss the visual system (defined previously) in relation to reading, attentional processes and Dyslexia.
- The authors need to introduce more clearly the relationship between their chosen tasks and how they differently tap into the Magno- and Parvo-cellular visual pathways. In doing so they should cite the relevant literature, for example: V. P., Nealey, T. A., & Maunsell, J. H. (1992). Mixed parvocellular and magnocellular geniculate signals in visual area V4. Nature, 358(6389), 756–761.
- To make the paper more accessible clarify the brain networks in the introduction, and specify brain regions involved. The discussion of the ROIs from the current study does not refer back to the brain networks and specific regions mentioned in the introduction. The main brain networks and attentional systems of interest should be defined in the introduction to facilitate the discussion of the ROIs and the brain networks in which they are involved.
Line 79: In the introduction the authors outline the hypothesis that abnormal input via the M system could cause dysfunction in the fronto-parietal attentional network (line 79) which is important for attentional shifting during letter and word processing. However, in the second paragraph of the discussion various networks are mentioned without clearly explaining how these relate to this hypothesis and the current results.
Line 467: ventral attentional network
Line 470: parieto-prefrontal network
Line 476: right frontal parietal system
Line 477: left frontal parietal system
- The univariate analysis should be explained prior to the MKL in the introduction-line 174
- The classification needs to be described prior to the description of how it was used in Section 2.7.
Minor Comments:
-In the methods, on line 261, please provide more details for the description of the “M stimuli” and “P stimuli”.
-Line 533-535: the authors state a limitation of the study was a small sample size but that it was within the range used of previous studies. This contradicts one of the motivations for the study outlined in the introduction (line 160-165) which was to test a larger sample size than previous studies. Perhaps in the discussion the authors could explain that although they tested a larger sample size than previous studies with dyslexic readers, the sample size is still relatively small.
-Line 543 and line 558-559: the authors seem to contradict themselves in the last two paragraphs which may take away from the message of the paper. I would suggest rephrasing the limitation of the study (line 543) so that it does not appear to go against the conclusion that the M pathway is “reliable biomarker of DD” (line 557-558).
-Figure 3 should clearly state the reading measure.
Style and language:
Line 63: delete “that”, “a phonological system thatrecognizes…”
Line 91: i.e. and e.g. should have a comma after it. Make changes throughout manuscript.
Line 93: respectively doesn’t make sense here, it can be deleted.
Line 57-60: the structure and use of parentheses in the sentence makes it difficult to read.
Line 108: add “to”, “have been widely employed toinvestigate…”
Line 139: semi-colon should be a period.
Lines 161-163: authors cited in text only in two out of three references.
Line 167: correct to “during development”.
Line 170: i.e. not needed.
Line 305: missing article, “in apseudorandom order”.
Line 312: “instead” doesn’t make sense here.
Line 319: missing “and” for final list item.
Line 436: weights
Line 517: change word order and delete comma, “the ROIs which lead classification in this task have the lowest…”
Author Response
Reviewer #1:
R: This study aims to characterize the brain networks that may be related to a deficit in the Magnocellular (M) system in dyslexic readers. To this end the authors employed stimuli that are known to activate dorsal and ventral visual streams which receive input from the M system and Parvocellular (P) systems respectively, using two tasks 1) full-field sinusoidal gratings and 2) motion coherence. They manipulated spatial and temporal frequencies, luminance contrast, as well as threshold and supra-threshold levels, in an attempt to design more sensitive tasks that assess the integrity of the dorsal and ventral visual streams within the M and P visual systems. While previous studies have employed univariate analyses to examine group differences in visual processing (between Dyslexic and typical readers), the current study employed whole-brain multivariate analyses to identify disparate attentional networks in the brain that may contribute to the visual M stream. They found that weaker neural activation along the dorsal visual stream and in areas within the fronto-parietal attentional network in dyslexic readers was able to discriminate between Dyslexic and typical reading controls.
Overall, the manuscript is well-written. The authors outline the theoretical and methodological motivations for their study well, they highlight how visual processing within the M system relates to reading specific processes, and they draw clear conclusions from their findings. The results contribute to research geared towards early identification of Developmental Dyslexia and possible approaches to remediation that focus on perceptual training targeting the dorsal visual stream.
Authors: We would like to thank the Reviewer 1 who formulated favorable comments on this manuscript and gave us important and constructive suggestions that helped us in refining it.
R: Major Comments:
- It would help readers to follow the argument of the manuscript if the magnocellular (M) system and dorsal and ventral visual streams were defined earlier in the introduction because the second paragraph is not clear. In the current version, it was also not clear that the dorsal and ventral streams receive different input from the M and P visual systems. I recommend reordering the structure of the second and third paragraphs as follows:
- Describe the two neurocognitive systems involved in reading.
- Define the M and P systems, and dorsal and ventral streams.
- Discuss the visual system (defined previously) in relation to reading, attentional processes and Dyslexia.
A: We thank Reviewer for this suggestion.
We re-ordered the structure of the second and third paragraph of the Introduction as follows (lines 73-119):
“There is now a great deal of evidence supporting the role of the visual magnocellular (M) system in reading [39]. Reading requires the integration of two distinct neurocognitive systems: a visual system recognizes a visual word form within a crowded group of letters, and a phonological language system rapidly recognizes and produces spoken words from a group of phonemes [40]. The M and parvocellular (P) pathways are major parallel visual system streams that, […].
Therefore, it is plausible to hypothesize that a weakened or abnormal M input in the dorsal visual stream, and consequent dysfunction of the main fronto-parietal attentional network, could be a neurobiological substrate of SAS in DD [53-54].”
We coherently re-numbered the references.
R: 2. The authors need to introduce more clearly the relationship between their chosen tasks and how they differently tap into the Magno- and Parvo-cellular visual pathways. In doing so they should cite the relevant literature, for example: V. P., Nealey, T. A., & Maunsell, J. H. (1992). Mixed parvocellular and magnocellular geniculate signals in visual area V4. Nature, 358(6389), 756–761.
A: We thank Reviewer for this comment.
We clarified the relationship between our tasks and the two visual pathways as follows (lines 120-125):
“Sinusoidal gratings and sensitivity for motion coherence are two classic psychophysical visual tasks [56] that have been widely employed to investigate the neural correlates of the dorsal and ventral visual pathways in the general population [e.g., 42-43,57-64] and in subjects with DD [44,58,65-67], as they tap the psychophysical characteristics of the two cell classes (i.e., M and P) in-volved in the two visual pathways [68].”
R: 3. To make the paper more accessible clarify the brain networks in the introduction, and specify brain regions involved. The discussion of the ROIs from the current study does not refer back to the brain networks and specific regions mentioned in the introduction. The main brain networks and attentional systems of interest should be defined in the introduction to facilitate the discussion of the ROIs and the brain networks in which they are involved.
A: We thank Reviewer for this suggestion.
We clarified the brain regions involved in the “reading circuit” in the Introduction (lines 53-64):
“After initial processing of print occurs within the left-hemispheric fusiform gyrus (the visual word form area), a large left hemisphere circuit including the supramarginal gyrus (orthography to phonology mapping), the superior temporal gyrus (phonological processing), the inferior parietal lobule and the angular gyrus (lexical-semantic processing), and the inferior frontal gyrus (phonological and semantic processing, working memory), is engaged [9-11]. Moreover, subcortical regions implicated in long-term and working memory, procedural learning and rapid sequential auditory processing (thalamus, basal ganglia and hippocampus), have also been implicated in reading [12-14]. Finally, left and right fronto-parietal network [15-16] strongly modulate both the visual and auditory word pathway by temporal and spatial selective attention [17].”
R: Line 79: In the introduction the authors outline the hypothesis that abnormal input via the M system could cause dysfunction in the fronto-parietal attentional network (line 79) which is important for attentional shifting during letter and word processing. However, in the second paragraph of the discussion various networks are mentioned without clearly explaining how these relate to this hypothesis and the current results.
Line 467: ventral attentional network
Line 470: parieto-prefrontal network
Line 476: right frontal parietal system
Line 477: left frontal parietal system
A: We thank Reviewer for this remark.
We made the terminology used in the Introduction and the Discussion sections uniform in order to explain how our findings relate to the hypothesis (lines 104-108):
“The visual M system contributes to promptly recognizing and sequencing letters. The former happens by rapidly focusing ventral attention network (VAN) on the letter to be identified, while the latter happens by logging the amplitude and the order of attentional shifts and eye movements during the inspection of each word [32-33,38,49-50].”
… and (lines 500-520):
“Overall, the multivariate approach significantly discriminated between TRs and children with DD. In particular, we demonstrated that functional activation in the entire visual dorsal stream and the VAN ranks highest (Table 2). Although none of the T-test would survive the multiple comparison correction, their qualitative analysis indicated that activation in discriminative ROIs is generally higher in TRs than in children with DD (Table 3) [44,58,65-67]. This pattern of results is consistent with the notions that the fronto-parietal network is linked to "dorsal stream vulnerability" underlying several neurodevelopmental disorders, and that the functionality of the M pathway is intimately related to the attention systems involved in attention control [51-52]. Deficits in the M pathway could influence higher visual processing stages through the dorsal stream, and, therefore, lead to reading difficulties through impaired attentional orienting [32,38,69]. While the right fronto-parietal system is a crucial component of the network subserving the automatic shifting of attention [15-16], the left fronto-parietal system has been linked to auditory word form processing [135]. Developmental changes in activation of the right fronto-parietal system have been linked to reading skills in both children with DD [136-138] and TRs [139]. Taken together, our findings suggest that a weakened or abnormal M input in the dorsal visual stream, and con-sequent dysfunction of the main fronto-parietal attentional network, is associated with SAS in DD [53-54].”
R: 4. The univariate analysis should be explained prior to the MKL in the introduction-line 174
A: We thank Reviewer for this comment.
We better introduced the univariate analysis in the Introduction (lines 160-163):
“Univariate analysis is the simplest approach for neuroimaging statistical analysis, as it deals with each region (or image voxel) independently from each other. Thus, it investigates just the direct relationship between DD and the single region, without considering the nature of interdependencies between brain regions underlying the visual M stream and the brain regions sensitive to the associated M demands [44,58,65-67].”
R: 5. The classification needs to be described prior to the description of how it was used in Section 2.7.
A: We thank Reviewer for this suggestion.
We moved the multivariate analysis breakdown from the beginning to the end of Section 2.7, after the description of the classification method we used (lines 433-441):
“To summarize, the whole multivariate analysis can be broken down into four steps: firstly, we performed a classification experiment to discriminate between TRs and children with DD by using whole brain ROIs. Secondly, we performed an analysis on the selected classifier to identify which ROIs and contrast maps contributed most to the classification. Thirdly, we ran univariate post-hoc analyses (i.e., T-tests) to investigate the direction and the magnitude of the differences between TRs and children with DD in the selected ROIs. Finally, we tested for correlations between task-induced cortical activation in the different ROIs and the neuropsychological tests.”.
R: Minor Comments:
-In the methods, on line 261, please provide more details for the description of the “M stimuli” and “P stimuli”.
A: We thank Reviewer for this remark.
We provided more details for the M and P stimuli in the ‘2.4.1. Full-field sinusoidal gratings [43]’ paragraph (lines 278-295):
“The M and P stimuli were designed to elicit differential bold responses from M and P pathways. The M stimulus was a monochrome, low spatial frequency, high temporal frequency, high luminance contrast, full-field sinusoidal grating with sinusoidal counterphase flicker; the P stimulus was a high color contrast, high spatial frequency, low temporal frequency, low luminance contrast full-field sinusoidal grating with sinusoidal counterphase flicker. The M stimulus was a 100% luminance contrast, black-white grating with a spatial frequency of 0.5 cycles per degree (cpd) and a flicker frequency of 15 Hz. The P stimulus was a low luminance contrast, high color contrast red–green grating with a spatial frequency of 2cpd and a flicker frequency of 5 Hz. Color levels in the P stimulus were set to be near-isoluminant, the red luminance was set to the maximum level and the green was set to 39% of the maximum level, as implemented in Denison and colleagues [43]. The blank stimulus was a gray screen of mean luminance. The outer borders of each stimulus faded into gray to avoid sharp visual edges at the stimulus boundaries. Both gratings were presented at one of 6 orientations (0°, 30°, 60°, 90°, 120°, 150°) and changed to the next orientation every 2.33s.”
R: -Line 533-535: the authors state a limitation of the study was a small sample size but that it was within the range used of previous studies. This contradicts one of the motivations for the study outlined in the introduction (line 160-165) which was to test a larger sample size than previous studies. Perhaps in the discussion the authors could explain that although they tested a larger sample size than previous studies with dyslexic readers, the sample size is still relatively small.
A: We thank Reviewer for this remark.
We re-phrased this limitation of our study as follows (lines 568-575):
“First, although we tested a larger sample size than those described in previous studies, the sample size of our study is still relatively small, which may have affected the classification performance and model generalizability [149]. Nonetheless, our sample was sufficient for a ranking profile of brain regions consistent with previous neuroimaging findings and etiological models of DD. It is plausible to predict that the greater power afforded by using larger samples would produce classification accuracy indices larger than those reported herein.”
R: -Line 543 and line 558-559: the authors seem to contradict themselves in the last two paragraphs which may take away from the message of the paper. I would suggest rephrasing the limitation of the study (line 543) so that it does not appear to go against the conclusion that the M pathway is “reliable biomarker of DD” (line 557-558).
A: We thank Reviewer for this suggestion.
We re-phrased this limitation of our study as follows (lines 579-582):
“Third, the implemented procedure selects the minimal number of ROIs that provide a significant contribution to the classification. Consequently, ROIs that give a minor contribution to the classification, but that are involved in the “reading network”, may not be highlighted.”
R: -Figure 3 should clearly state the reading measure.
A: We thank Reviewer for this remark.
We stated the reading measure in the caption of Figure 3 (lines 639-640):
“Reading represents a mean score of text [101-102], single unrelated words and pronounceable pseudo-words [103-104] reading tests.”
R: Style and language:
Line 63: delete “that”, “a phonological system that recognizes…”
A: Done.
R: Line 91: i.e. and e.g. should have a comma after it. Make changes throughout manuscript.
A: Done.
R: Line 93: respectively doesn’t make sense here, it can be deleted.
A: Done.
R: Line 57-60: the structure and use of parentheses in the sentence makes it difficult to read.
A: We thank Reviewer for this remark.
We re-phrased this sentence as follows (lines 69-72):
“(…) other theoretical models investigating deficits in underlying sensory (i.e., deficits in the auditory system [20-30]) and cognitive mechanisms (i.e., deficits in the rapid automatized naming [5,31] and in visual and auditory attention [32-38]) that potentially cause DD remain compelling.”
R: Line 108: add “to”, “have been widely employed to investigate…”
A: Done.
R: Line 139: semi-colon should be a period.
A: Done.
R: Lines 161-163: authors cited in text only in two out of three references.
A: We thank Reviewer for this remark.
According to the author guidelines, we adjusted the references in this sentence (lines 179-184):
“Third, previous studies have had relatively small samples (adults with DD n=6 and adult TRs n=8 [44]; adults with DD n=5 and adult TRs n=5 [65-66]; children with DD n=14, age-matched TRs n=14 and reading-matched TRs n=10 [67]), and few neuroimaging studies have directly investigated group differences in the visual M pathway in children with DD [58,67].”
R: Line 167: correct to “during development”.
A: Done.
R: Line 170: i.e. not needed.
A: Done.
R: Line 305: missing article, “in a pseudorandom order”.
A: Done.
R: Line 312: “instead” doesn’t make sense here.
A: Done.
Line 319: missing “and” for final list item.
A: Done.
R: Line 436: weights
A: Done.
R: Line 517: change word order and delete comma, “the ROIs which lead classification in this task have the lowest…”
A: Done.
Reviewer 2 Report
According to the magnocellular theory of developmental dyslexia, a deficit in the visual magnocellular system leads to a dysfunction of the fronto-parietal attentional network. The magnocellular theory of developmental dyslexia is an intriguing theory to explain at least some features of developmental dyslexia. Given the pros and cons, the theory is still a matter of debate. In this situation, it is desirable that new studies look for more evidence for the correctness of this theory. In the present study, the authors performed a functional MRI analysis investigating regions of interests (ROIs) of the brains of dyslexic children and normal readers. The results were compared using a multivariate pattern analysis.
Introduction
The introduction is clearly written and gives a lucid introduction to the scientific background and the objectives of the study.
Methods
MRI methods and data analysis are clearly described. However, since not all readers are familiar with MRI analysis I suggest to explain the meaning of ROI in a few words, e. g. „a method to extract data from a subset of voxels using an atlas that partitions the brain in anatomical regions of interest.“ As it is difficult to recruit children without neurological abnormalities for an MRI examination, a group of 25 children with dyslexia and 24 normal readers can already be considered a large group.
Results
Lines 421-422: The authors write that „…none of the T-test survived a multiple comparison correction …“ and that „… the analysis of the T-values provides useful information about the group differences.“ If the Bonferroni corrected p-values of a T-test don´t reach significance, there is no evidence for group differences.
Lines 423-424: Even if activation in discriminative ROIs is higher in normal readers than in children with dyslexia this does not mean that this is evidence for a scientific result as long as the p-value is not sufficiently small.
Lines 425-429 and Supplementary Table 2: Values of p<0.05 that are not Bonferroni corrected also cannot be considered as scientific evidence of a difference. Therefore, the authors should clearly state that in their studies there is no evidence for a difference between the threshold CM contrasts and all other contrasts. Even if a Bonferroni corrected p-value of p<0.05 is obtained, it is questionable to interpret such a result as scientific evidence. Although this corresponds to an arbitrary labeling as "significant", it is statistically challengeable (e. g. Joannidis: PLOS Medicine, 2005; Wasserstein et al.: The American Statisticion, 2019; Billheimer: The American Statisticion, 2019).
Lines 444- 451: A positive correlation was only found between activation in area 7P and 7A when testing with coherent motion level 6% contrast maps vs baseline. Here p-values of p=0.0001 and p=0.0002 suggest that this can be regarded as a positive result. Hyperactivation in these areas correlated with better reading performance. It is, however, difficult to interpret this result in favor of the magnocellular theory of developmental dyslexia as it is unclear what role this hyperactivation plays in the reading process. Could the authors please describe in more detail the different performance of children with dyslexia and normal readers on the subtests of the reading test.
Discussion
Accordingly, if the p-values do not allow to detect a difference between groups, the results should not be discussed as positive findings, but it should be made clear that the results do not support the magnucellular theory of developmental dyslexia.
Conclusions
Lines 557-564: The authors claim that the „… results further support the M visual pathway as a reliable biomarker of DD [31,143], which could lead to new approaches in the diagnosis of this neurodevelopmental disorder [144]. Moreover, these findings pave the way for the creation of early identification protocols, more effective prevention programs and better -defined rehabilitative treatments.“ However, given the results of the present study, there is no basis for this claim.
From my point of view, the result that no differences were found between activation in ROIs of children with delelopmental dyslexia and normal readers is quite interesting and worth publishing regardless of whether the result supports the magnocellular theory of developmental dyslexia.
Author Response
Reviewer #2:
R: According to the magnocellular theory of developmental dyslexia, a deficit in the visual magnocellular system leads to a dysfunction of the fronto-parietal attentional network. The magnocellular theory of developmental dyslexia is an intriguing theory to explain at least some features of developmental dyslexia. Given the pros and cons, the theory is still a matter of debate. In this situation, it is desirable that new studies look for more evidence for the correctness of this theory. In the present study, the authors performed a functional MRI analysis investigating regions of interests (ROIs) of the brains of dyslexic children and normal readers. The results were compared using a multivariate pattern analysis.
Introduction
The introduction is clearly written and gives a lucid introduction to the scientific background and the objectives of the study.
A: We would like to thank the Reviewer 2 who formulated favorable comments on this manuscript and gave us important and constructive suggestions that helped us in refining it.
R: Methods
MRI methods and data analysis are clearly described. However, since not all readers are familiar with MRI analysis I suggest to explain the meaning of ROI in a few words, e. g. „a method to extract data from a subset of voxels using an atlas that partitions the brain in anatomical regions of interest.“ As it is difficult to recruit children without neurological abnormalities for an MRI examination, a group of 25 children with dyslexia and 24 normal readers can already be considered a large group.
A: We thank Reviewer for these comments.
We introduced the concept of the ROI approach in the ‘2.6. fMRI data processing (see Supplementary Figure 1)’ paragraph (lines 368-376):
“Finally, each contrast map was partitioned into Regions of Interest (ROI), an approach that extract data from a subset of voxels belonging to a homologous anatomo-functional brain region improving the signal to noise ratio. In particular, the mean value of each contrast map was computed for each subject in the cortical ROI from the HCP-MMP1 atlas by using a linear model that al-lowed to control for the effects of performance IQ and DSM-IV-I in order to remove the amount of the signal explained by these confounders from our de-pendent variable (i.e., the mean value of each contrast map).”
Moreover, we re-phrased the limitation of our study regarding sample size as follows (lines 568-575):
“First, although we tested a larger sample size than those described in previous studies, the sample size of our study is still relatively small, which may have affected the classification performance and model generalizability [149]. Nonetheless, our sample was sufficient for a ranking profile of brain regions consistent with previous neuroimaging findings and etiological models of DD. It is plausible to predict that the greater power afforded by using larger samples would produce classification accuracy indices larger than those reported herein.”
R: Results
Lines 421-422: The authors write that „…none of the T-test survived a multiple comparison correction …“ and that „… the analysis of the T-values provides useful information about the group differences.“ If the Bonferroni corrected p-values of a T-test don´t reach significance, there is no evidence for group differences.
R: Lines 423-424: Even if activation in discriminative ROIs is higher in normal readers than in children with dyslexia this does not mean that this is evidence for a scientific result as long as the p-value is not sufficiently small.
R: Lines 425-429 and Supplementary Table 2: Values of p<0.05 that are not Bonferroni corrected also cannot be considered as scientific evidence of a difference. Therefore, the authors should clearly state that in their studies there is no evidence for a difference between the threshold CM contrasts and all other contrasts. Even if a Bonferroni corrected p-value of p<0.05 is obtained, it is questionable to interpret such a result as scientific evidence. Although this corresponds to an arbitrary labeling as "significant", it is statistically challengeable (e. g. Joannidis: PLOS Medicine, 2005; Wasserstein et al.: The American Statisticion, 2019; Billheimer: The American Statisticion, 2019).
A: We thank Reviewer for these comments.
Univariate post-hoc analyses (i.e., T-tests) were performed to independently investigate the direction and magnitude of the differences between TRs and children with DD for each contrast in the highest performance ROIs in the significant model resulted by the multivariate analysis. Although none of the T-test survived a whole-brain multiple comparison correction, the qualitative interpretation of the T-values provided useful information about the group differences, which were complementary to the significant results obtained with the multivariate analysis. Moreover, some p-values can be considered sufficiently small (i.e., ≤0.001 uncorrected) and can provide useful information for qualitatively commenting our findings (Table 3).
We better specified the steps of the multivariate analyses (lines 424-441):
“As contrast weight analysis is influenced by the complementary information provided by the different contrasts in the selected ROIs, we performed univariate post-hoc analyses (i.e., T-tests) to independently investigate the direction and magnitude of the differences between TRs and children with DD for each contrast.
(…) Thirdly, we ran univariate post-hoc analyses (i.e., T-tests) to investigate the direction and the magnitude of the differences between TRs and children with DD in the selected ROIs. Finally, we tested for correlations between task-induced cortical activation in the different ROIs and the neuropsychological tests.”
(…) we re-phrased the ‘Results’ (lines 459-471):
“Univariate post-hoc statistical analyses at ROI level for each contrast are reported in Table 3. Although none of the T-test survived a whole-brain multiple comparison correction, the qualitative interpretation of the T-values provides useful information about the group differences, which are complementary to the significant results obtained with the multivariate analysis. Almost all T-values were positive, indicating that activation in discriminative ROIs is generally higher in TRs than in children with DD (Table 3). Furthermore, the magnitude of the T-values for threshold CM contrasts (i.e., CML6-vs-B and CML15-vs-B) and for the M-vs-B contrast, showed on average larger differences between the two groups compared to the other contrasts (Table 3). Interestingly, the ROIs with the largest differences between the two groups show higher contrast weights also in the multivariate analysis (Table 2).”
(…) and we added the p-values in Table 3:
R: Lines 444- 451: A positive correlation was only found between activation in area 7P and 7A when testing with coherent motion level 6% contrast maps vs baseline. Here p-values of p=0.0001 and p=0.0002 suggest that this can be regarded as a positive result. Hyperactivation in these areas correlated with better reading performance. It is, however, difficult to interpret this result in favor of the magnocellular theory of developmental dyslexia as it is unclear what role this hyperactivation plays in the reading process. Could the authors please describe in more detail the different performance of children with dyslexia and normal readers on the subtests of the reading test.
A: We thank Reviewer for this comment.
We described the differences in reading performance of children with DD and TRs by explaining that children with DD showed significantly lower scores in all reading tests compared to TRs (lines 247-248):
“As expected, children with DD showed significantly lower scores in all reading tests compared to TRs”
R: Discussion
Accordingly, if the p-values do not allow to detect a difference between groups, the results should not be discussed as positive findings, but it should be made clear that the results do not support the magnucellular theory of developmental dyslexia.
A: We thank Reviewer for this remark.
The multivariate analysis with the GL-MKL method resulted in a model which significantly discriminates between TRs and children with DD with 65.9% accuracy and 64.7% AUC (p-value=0.043), and identified a set of 11 highly discriminative ROIs within the entire visual dorsal stream and the VAN (Figures 1-2). These brain areas lie within the fronto-parietal network which is linked to the "dorsal stream vulnerability" underlying several neurodevelopmental disorders, and are known to be involved in deficits in the SAS in DD. As previously explained, univariate analyses (i.e., T-tests) were performed to independently investigate the direction and magnitude of the differences between TRs and children with DD for each contrast in these ROIs. Their qualitative interpretation provided useful information about the group differences, which were complementary to the significant results obtained with the multivariate analysis.
We re-phrased Discussion as follows (lines 493-567):
“(…). The multivariate approach performed by using the MKL-ROI classifier allows us to explore the widespread effects of the diagnosis of DD across multiple brain ROIs. Thus, it is more suitable to investigate the neural correlates of neurodevelopmental disorders involving wide brain networks compared to univariate analysis, aimed to test the relationship between the diagnosis of DD and each single ROIs.
Overall, the multivariate approach significantly discriminated between TRs and children with DD. In particular, we demonstrated that functional activation in the entire visual dorsal stream and the VAN ranks highest (Table 2). Although none of the T-test would survive the multiple comparison correction, their qualitative analysis indicated that activation in discriminative ROIs is generally higher in TRs than in children with DD (Table 3) [44,58,65-67].
(…)
Regarding the full-field sinusoidal gratings, the ROIs which lead classification in this task have the lowest ROI weights in the MKL classifier (Table 2). Although the M-vs-B and P-vs-B contrast weights are comparable within these ROIs, the M-vs-B T-values are often higher in magnitude compared to the P-vs-Bs (Table 3). So, as has been hypothesized for the CM, it is likely that the comparison between M and P stimuli could be relevant to the classification of TRs and children with DD. These findings further suggest that the M pathway is impaired in individuals with DD, whereas the other major parallel pathway of the visual system, the parvocellular stream, is less severely or not at all affected [38-39,54,69].”
R: Conclusions
Lines 557-564: The authors claim that the „… results further support the M visual pathway as a reliable biomarker of DD [31,143], which could lead to new approaches in the diagnosis of this neurodevelopmental disorder [144]. Moreover, these findings pave the way for the creation of early identification protocols, more effective prevention programs and better -defined rehabilitative treatments.“ However, given the results of the present study, there is no basis for this claim.
From my point of view, the result that no differences were found between activation in ROIs of children with delelopmental dyslexia and normal readers is quite interesting and worth publishing regardless of whether the result supports the magnocellular theory of developmental dyslexia.
A: We thank Reviewer for this comment.
As previously explained, the multivariate analysis with the GL-MKL method resulted in a model which significantly discriminates between TRs and children with DD with 65.9% accuracy and 64.7% AUC (p-value=0.043), and identified a set of 11 highly discriminative ROIs within the entire visual dorsal stream and the VAN (Figures 1-2). The qualitative interpretation of the univariate post-hoc analyses (i.e., T-test) indicated that activation in these discriminative ROIs is generally higher in TRs than in children with DD (Table 3). This pattern of results is consistent with the notions that the fronto-parietal network is linked to "dorsal stream vulnerability" underlying several neurodevelopmental disorders, and that the functionality of the M pathway is intimately related to the attention systems involved in attention control.
We better explained our results in the Discussion (lines 493-520):
“The multivariate approach performed by using the MKL-ROI classifier allows us to explore the widespread effects of the diagnosis of DD across multiple brain ROIs. Thus, it is more suitable to investigate the neural correlates of neurodevelopmental disorders involving wide brain networks compared to univariate analysis, aimed to test the relationship between the diagnosis of DD and each single ROIs.
Overall, the multivariate approach significantly discriminated between TRs and children with DD. In particular, we demonstrated that functional activation in the entire visual dorsal stream and the VAN ranks highest (Table 2).
(…) Taken together, our findings suggest that a weakened or abnormal M input in the dorsal visual stream, and consequent dysfunction of the main fronto-parietal attentional network, is associated with SAS in DD [53-54].”
(…) and in the Conclusions (lines 584-589):
“The MKL-ROI approach used in the present study identified a highly dis-criminative network in the entire visual dorsal stream and the VAN when comparing children with DD and age-matched TRs. These brain areas lie within the fronto-parietal network which is linked to the "dorsal stream vulnerability" underlying several neurodevelopmental disorders, and are known to be involved in deficits in the SAS in DD.”
